# Organ-Specific Glucose Uptake: Does Sex Matter?

**DOI:** 10.3390/cells11142217

**Published:** 2022-07-16

**Authors:** Adithi Gandhi, Ryan Tang, Youngho Seo, Aditi Bhargava

**Affiliations:** 1Center for Reproductive Sciences, Department of Obstetrics and Gynecology, University of California San Francisco, San Francisco, CA 94143, USA; adithigandhi2022@u.northwestern.edu; 2Department of Radiology and Biomedical Imaging, University of California San Francisco, San Francisco, CA 94143, USA; ryantng07@berkeley.edu (R.T.); youngho.seo@ucsf.edu (Y.S.)

**Keywords:** brown fat, female, ^18^F-fluorodeoxyglucose, heart, high-fat diet, male, PET scan, sex differences, skeletal muscle, young mice

## Abstract

Glucose uptake by peripheral organs is essential for maintaining blood glucose levels within normal range. Impaired glucose uptake is a hallmark of type 2 diabetes (T2D) and metabolic syndrome and is characterized by insulin resistance. Male sex is an independent risk factor for the development of T2D. We tested whether sex and diet are independent variables for differential glucose uptake by various organs. Here, in a longitudinal study, we used ^18^F-fluorodeoxyglucose (FDG) and positron emission tomography (PET) to determine baseline differences in whole-body glucose uptake in young male and female mice on chow and high-fat diets. We report that sex and diet are important independent variables that account for differential glucose uptake in brown fat, skeletal muscle, liver, heart, kidney, and the stomach, but not the brain, lungs, pancreas, small intestine, or perigonadal adipose. Of the seven organs analyzed, two organs, namely brown fat, and the heart had the highest concentrations of FDG, followed by the brain, kidneys, and skeletal muscle on chow diet. Young female mice had 47% greater FDG uptake in the brown fat compared to male mice, whereas skeletal muscle FDG uptake was 49% greater in male mice. The high-fat diet inhibited FDG uptake in brown fat, skeletal muscle, and the heart, three major organs involved in uptake, whereas brain uptake was enhanced in both sexes. These foundational and groundbreaking findings suggest that mechanisms of glucose homeostasis are context- and organ-dependent and highlight the need to study sex-specific outcomes and mechanisms for diseases such as T2D, obesity, and metabolic syndrome.

## 1. Introduction

Glucose uptake by different organs is key for maintaining homeostatic blood glucose levels. It is well documented that insulin and glucagon are essential to regulating glucose uptake. Insulin, a peptide hormone, is secreted by pancreatic beta cells in response to increased blood glucose levels and activates glucose transporters that allows for the uptake of glucose into various organs [1]. Glucagon, another peptide hormone, also secreted by the pancreas, acts directly on the liver to stimulate glucose production, and decrease glucose uptake [2]. 

Blood glucose levels are maintained by coordinated uptake by various organs, but the contribution of each organ in maintaining homeostasis in males versus females is not known. Besides compromised action of insulin and other peptide hormones, ineffective glucose uptake contributes to increased levels of blood glucose, resulting in health issues such as type 2 diabetes, metabolic syndrome, retinopathy, and neuropathy [3]. Thus, it is critical to understand how glucose uptake varies by organ. 

Most studies focus on the role of glucose uptake in skeletal muscle and liver. In the liver, glucose is mostly stored in the form of glycogen [4]. The hepatic glycogen is the major source of glucose under fasting conditions; to meet the energy demands of various organs under fasting conditions, glycogen is converted to glucose via glycogenolysis and thereby helps regulate blood glucose levels [4]. Around 80% of insulin-stimulated glucose uptake reportedly occurs in skeletal muscle and is mediated by three GLUT (GLUT1, GLUT3, and GLUT4) transporters, mainly GLUT 4 [5]. Glucose uptake in skeletal muscle is also influenced by exercise since muscle contraction facilitates glucose uptake by catalyzing the transfer of GLUT transporters to the plasma membrane [5].

While insulin-stimulated glucose uptake in these organs has been extensively studied, not much is known about uptake in other organs, such as the pancreas, stomach, and kidneys, at basal state. Specifically, there have not been many studies investigating how glucose uptake varies by organ on a whole-body level, and rather most studies focus on organs such as the liver or skeletal muscle. 

Studies have shown that mechanisms of glucose uptake differ in disparate organs. For example, in the liver, glucose uptake occurs mainly through the GLUT2 transporter, where it is then phosphorylated by glucokinase to be stored as glycogen, using glycogen synthase [5]. In skeletal muscle, on the other hand, glucose uptake is facilitated mostly by the GLUT4 transporter, which translocates to the plasma membrane in response to urocortin 2 [6] or insulin, and it is then phosphorylated and stored as glycogen there, as well [7]. However, the mechanisms and contribution of other organs in glucose uptake is understudied or unknown. Before mechanisms can be delineated, it is important to first establish how glucose uptake varies by organ. 

Sex is suggested to be an important factor in insulin resistance and diabetes prevalence. Studies have indicated that type II diabetes (T2D) seems to be more prevalent in men, specifically when compared to premenopausal women [8]. In experimental models, estrogens may confer a protective role in maintaining insulin sensitivity in female rats [9]. Indeed, insulin sensitivity for healthy individuals tended to be higher in women due to enhanced uptake in skeletal muscle [10]. For example, one study investigating sex differences in a high-fat diet-induced diabetes model found an interaction between diet and insulin levels modulated by the stress receptor *Crhr2* [11]. Therefore, sex differences in glucose uptake are important to understand. Diet has also been known to alter glucose uptake. Specifically, various animal models investigating the impacts of a high-fat diet have found that it negatively impacts glucose uptake [12,13,14]. However, there has not been much research into the interaction between sex and diet for glucose uptake in various organs. 

Brown adipose tissue has been noted to be important in glucose uptake and thus has been an area of interest for potential therapies for metabolic disorders [15]. It has been suggested that there are two pathways for uptake. The first is the typical insulin-stimulated pathway that works through the GLUT 4 receptor and the phosphoinositide 3-kinase–phosphoinositide-dependent kinase-1–Akt (PI3K–PDK1–Akt) pathway [6,15]. The second has not been rigorously studied but occurs during thermogenesis, such as in the brown fat, using energy to produce heat, and may involve the translocation of GLUT1 to the plasma membrane via an mTOR complex [15]. In humans, overall, brown fat tends to be more prevalent in younger babies and lean adults and has been suggested to play a role, in maintaining blood glucose levels and guarding against insulin resistance [16]. 

This study measured longitudinal whole body glucose uptake using ^18^F^-^fluorodeoxyglucose (FDG) and positron emission tomography (PET) in live male and female mice first on a standard chow diet followed by imaging on a high-fat diet (HFD).

## 2. Materials and Methods

### 2.1. Animals

All animal procedures were approved by the Institutional Animal Care and Use Committee (IACUC) at the University of California San Francisco and were conducted in accordance with the National Institutes of Health Guide for the Care and Use of laboratory Animals. Approved animal study numbers were AN181830 and AN191219. Male and female littermates (C57BL/6) bred at UCSF animal facility, at 12 weeks of age, were used. Mice were group housed and provided with enrichment. At the end of the experiments, mice were euthanized, and organs such as the pancreas, stomach, small intestine, and the gonadal adipose tissues were additionally collected for measurement of glucose uptake.

### 2.2. Diets

The mice were fed with a standard chow diet that consisted of 9% fat (PicoLab mouse Diet 20 #5058, Lab Supply, Fort Worth, TX, USA). Subsequently, after imaging for the chow diet, mice were switched to a 60% kcal high-fat diet (HFD) (Catalog# 12492 Research Diets, NJ, USA) for 2 weeks before re-imaging for FDG uptake. 

### 2.3. Positron Emission Tomography (PET)

In order to minimize diet-associated competitive inhibition of FDG uptake, animals were fasted for 5–6 h before intravenous administration of FDG. Mice were weighed before injection of FDG on each diet (Appendix A). Approximately 3.7 MBq of FDG in 0.1 mL was administered via the tail vein, using a custom-made catheter. After 55 min of uptake time, mice were scanned for 10 min, followed by microCT for attenuation correction and anatomical reference by microPET/CT (Inveon, Siemens Medical Solutions, Malvern, PA, USA). The parameters for microCT were 80 kVp X-ray tube voltage, 0.5 mA tube current, 175 ms exposure time per step, and 120 steps over 220°. During the uptake time, the animals were awake and kept warm on a temperature-controlled heating pad at 37°. The animals were under anesthesia with 2–2.5% isoflurane during the scan. Up to four animals were placed in the scanner each time, using a multi-animal bed. After the list mode data for PET were acquired, the manufacturer-provided reconstruction algorithm was used to reconstruct PET images with CT-based attenuation correction. PET and CT images were co-registered by using a precalculated transformation matrix to produce volumetric images in 128 × 128 × 159 matrices with a voxel size of 0.776 mm × 0.776 mm × 0.796 mm for PET, and 512 × 512 × 700 with a voxel size of 0.191 mm × 0.191 mm × 0.191 mm.

### 2.4. Volume of Interest (VOI) Analysis

For the volume of interest (VOI) analysis of FDG–PET images, an open-source image analysis software, Amide (amide.sourceforge.net), was used [17]. The standardized uptake value (SUV) normalized by body weight (activity concentration (Bq/mL) divided by injected activity (Bq) and then multiplied by body weight (g)—activities are synchronously decay corrected to either scan start time or injected time) was calculated for each organ VOI [18]. Spherical VOIs were placed for lungs, liver, left and right kidneys, skeletal muscle surrounding the femur, and brown fat behind the neck, and elliptical cylinder VOIs were placed for the brain and the heart. The dimensions of the VOIs were 5 mm long axis, 3 mm short axis, and 5 mm height for the brain; 1.5 mm short and long axes, and 3 mm height for the heart over the myocardium; 3 mm diameter for brown fat; and 2 mm diameter for all the rest. The VOI size and placement were consistent for all animal data. The difference in SUVs between sexes or diets were generated by maximum intensity projection and scaled the same for comparison.

### 2.5. Organ Counting 

Since VOIs could not be reliably computed for the pancreas, stomach, small intestine, and adipose fat, after the second round of imaging, the animals were euthanized, and these organs of interest were harvested. They were weighed and counted by using a gamma counter (HIDEX Automatic Gamma Counter, Turku, Finland). The %ID/g (percent of injected dose per gram) was derived for all counted organs.

### 2.6. Statistical Analysis

Sex and diet were the two independent variables of interest. Both the VOI and SUV data were analyzed by using a two-way analysis of variance (ANOVA) to investigate the main effects, followed by Mann–Whitney post hoc analysis when main effects were significant. A *p*-value > 0.05 was considered significant. For significant *p*-values, percent change (for comparisons between diet) and percent differences (for comparisons between sex) were also calculated.

## 3. Results

### 3.1. Sex Is an Important Determinant of Relative Glucose Uptake by Various Organs

Male sex is a known risk factor for type 2 diabetes, but the relative contribution of glucose uptake by different organs in males versus females has never been examined. All organs/tissues need glucose for survival; however, surprisingly, no systematic study exists that has examined the relative contribution of major organs/tissues at steady state in glucose uptake in males versus females. To ascertain the contribution of sex as an independent variable, we first determined whole-body FDG uptake on chow diet. The PET/CT hybrid imaging modality permits the simultaneous collection of molecular and morphologic information. We ascertained volumetric uptake in seven major organs involved in glucose uptake, namely the skeletal muscle, brain, kidney, liver, lungs, heart, and the brown fat. In addition, organ counting was performed on organs such as the stomach, small intestine, adipose, and the pancreas, for which volumetric analyses do not provide reliable data.

Skeletal muscle and hepatic tissues are thought to play a major role in maintenance of post-prandial glucose levels in the normal range and are the focus of majority of the studies. The volumetric analysis of region of interest (VOI) revealed that of the seven organs analyzed, two organs, namely brown fat, and the heart had the highest FDG SUV concentrations, followed by the brain, kidneys, skeletal muscle, liver, and lungs on chow diet (Figure 1A). We found that, under baseline physiological conditions, sex was an independent variable contributing to FDG uptake in several organs (Figure 1B–G), with the exception of the heart and brain (Figure 1C,F, respectively) and the lungs (Appendix A). The brown fat FDG uptake was 47% greater in young female mice versus male mice (Figure 1B). The skeletal muscle and kidney FDG uptakes were 49% and 17% more, respectively, in young male mice versus female mice (Figure 1E,G). 

### 3.2. Diet and Sex Are Important Determinant of Hepatic Glucose Uptake

Next, to determine if high-fat diet (HFD), a risk factor for diabetes, altered glucose uptake, we performed a longitudinal study. After determining whole-body FDG uptake on chow, mice of both sexes were fed a high-fat diet. The FDG uptake was 47% more in the brown fat of female mice compared with male mice (Figure 2A), and the HFD also decreased FDG uptake in brown fat in both sexes (Figure 2A). In diabetes, obesity, and metabolic syndrome, skeletal muscle and hepatic glucose uptake are reduced [19,20,21]. In agreement with these observations, the HFD decreased FDG uptake by ~70% in skeletal muscles in male mice and by 60% in female mice (Figure 2B and Appendix A). However, sex and diet were independent variables, and no interaction was found for FDG uptake in most organs examined (Figure 2A–F), except for the liver (Figure 2C). We find that the relative contribution of brown fat in FDG uptake is significantly more than skeletal muscle in female mice. The HFD was an independent variable contributing to the FDG uptake in the brown fat (Figure 2A), skeletal muscle, liver, heart, and the brain (Figure 2B–G), with only hepatic FDG exhibiting an interaction between sex and diet (Figure 2C). Surprisingly, diet did not influence FDG uptake in the kidneys (Figure 2E) or the lungs (Appendix A) in either sex, whereas the HFD significantly increased FDG uptake in the brains of female, and trended towards increased uptake in male mice, but did not reach significance (Figure 2F).

### 3.3. Stomach Glucose Uptake Differs between the Sexes

Bariatric surgery in obese people, regardless of biological sex, almost immediately resolves type 2 diabetes symptoms, with blood glucose levels being restored to normal range even before patients are discharged from the hospital. While changes in hormones are at play after surgery, it is not known if the baseline gut glucose uptake differs between males and females. FDG uptake in the gut (stomach and intestines), along with other soft tissues, such as the pancreas and adipose (perigonadal and mesenteric fats), is better quantified by using tissue counts rather than VOI. In our longitudinal study, mice on the high-fat diet showed significant differences in FDG uptake in the adipose and the stomach, with ~38% more uptake in the stomachs of female mice compared with male mice (Figure 3). Surprisingly, FDG uptake in the small intestine (along with mesenteric fat) or the pancreas did not differ between the sexes (Figure 3).

## 4. Discussion

Globally, 14 million more men are diagnosed with diabetes than women; sex differences in glucose clearance [22], glucose-stimulate insulin secretion, and glucagon-like peptide 1 are also reported [23]. Diabetes mellitus or T2D is comorbid with several pathologies that include metabolic syndrome, cardiovascular, and liver diseases. Most studies, whether using animal models of diabetes or in human subjects, assess either hepatic glucose uptake or skeletal muscle or both. No study thus far has considered that glucose uptake in either of these organs or others can vary by sex or be influenced by diet, or both. In humans, other confounders can be race, ethnicity, and physical exercise. In this study, we first examined FDG uptake in young male and female mice at a steady state. We found both sex and diet to be independent variables that influenced glucose uptake in several organs. Amongst the organs examined, liver was the only organ where sex and diet showed interaction for FDG uptake.

In fasted state, when blood glucose levels are low, the liver, kidney, and small intestine make glucose via a process called gluconeogenesis to keep up with the demand [24,25], whereas fat and muscle regulate gluconeogenesis [26,27]. Under conditions of fasting stress, glucose is synthesized (via glycogenolysis) in the liver and muscles. Insulin resistance is seen when liver, fat, and muscles do not respond well to endogenously secreted insulin and are unable to effectively take up glucose from blood (Figure 4). In diabetic individuals and those suffering from metabolic syndrome, this interplay between insulin and other regulatory hormones is lost; β-cells are unable to produce sufficient insulin levels to keep up with the demand, resulting in overt hyperglycemia.

Metabolic syndrome is ~3× more common than diabetes (prevalence is estimated at >1 billion people worldwide). Metabolic syndrome is a cluster of diseases such as hypertension, insulin resistance, and dyslipidemia. Metabolic syndrome increases the risk for other chronic diseases such as cardiovascular disease, chronic kidney disease, several types of cancers, and early death [28].

Meta-analyses reveal that diabetic women are at 50% greater risk for cardiovascular-related mortality and 30% greater risk for stroke than diabetic men. In the ADVANCE-On trial [29], gliclazide successfully controlled blood glucose levels in men, but not women. The use of sodium/glucose cotransporter 2 inhibitors resulted in a higher risk of genitourinary tract infections and ketoacidosis in women vs. men [30]. Women also have a higher risk of experiencing severe hypoglycemia with insulin therapy [31]. Female sex, statin use, smoking, diabetes duration, body weight, and initial HbA1c all predicted earlier failure of any dual treatment [22,32]. Rimonabant, an anti-obesity drug, was withdrawn from the market due to severe side effects in women [32]. Despite this mounting evidence of sex-differences in manifestation of metabolic disease in humans, very little is known about the sex differences in metabolic signaling pathways or the role of different organs in blood glucose uptake.

While skeletal-muscle glucose uptake is important and most studied for pathologies such as diabetes and metabolic diseases, brown fat is best known for its role in maintaining a lean phenotype. In humans, babies normally have significantly more brown fat than adults; it is primarily located behind the shoulder blades or the clavicles. Lean adults have more dense brown fat than obese or heavy adults [33]. Sex differences are reported in the amount of brown fat in humans; women reportedly store more brown fat depots than men and do not just have larger sized brown fat tissue [34]. Additionally, in rodents, non-shivering thermogenesis through brown fat is more efficient in females than in males [34]. The action of uncoupling proteins in the brown fat is key for energy expenditure, and glucose is a major source of energy that generates ATP once it enters glycolysis. We find that young-adult female mice demonstrate nearly 10% more FDG uptake by their brown fat than young-adult male mice after normalizing for both organ and body weights. Our data imply that, in females, brown fat, and not the skeletal muscle, might play a more key and significant role in pathologies related to impaired glucose, such as obesity, diabetes, and metabolic syndrome. In contrast to the skeletal muscle wasting that can be controlled via exercise, the processes and mechanisms that regulate loss and/or maintenance of brown fat, as we age, are unknown. It is possible that exercise and/or temperature may stimulate or maintain brown fat. In mice, extreme and unnatural stressors such as housing at cold temperatures (~4 °C or less) for extended periods of time can result in the conversion of white fat into beige fat [35] in appearance and increases expression of uncoupling protein, but the cold stress-mediated increase is transient, and its functional significance is unclear. Humans that live in cold temperatures, such as the Inuit Indians (Eskimos), do not reportedly have more brown fat than people who live in temperate climates; thus, stressors such as cold temperature may transiently increase the activity of brown fat, and this, in turn, might help with glucose clearance due to increased uptake, but the effects may not be sustained. Indeed, one study found that FGD uptake in brown fat increased in female and male adults upon exposure to cold (19 °C) [36].

FDG uptake by heart in both sexes was comparable at basal state, and the high-fat diet decreased uptake similarly in female and male mice. In the kidneys, FDG uptake was comparable between the sexes, although a trend toward enhanced uptake after the high-fat diet was seen in young female mice, whereas FDG uptake was lower in young male mice. It remains to be determined whether, as animals age, there is a differential FDG uptake by these organs. Surprisingly, no sex differences were noted in organs such as the pancreas or the small intestine, whereas the perigonadal adipose tissue exhibited more uptake in females. Adipose tissue is also an ectopic site for production of insulin and development of insulin resistance; however, in this study, we did not measure insulin stimulated FDG uptake or ascertain the changes in the expression levels of various GLUTs in different organs. High-fat diet increased FDG uptake in the brain. High glucose levels in the brain underlie many neurological pathologies that include dementia.

Relative insulin deficiency and insulin resistance are two hallmarks of diabetes. While insulin resistance is always present in the early phase, β-cell failure and glucose uptake determine the pace of diabetes onset. Sex differences in β-cell function are noted in clinical studies. Despite healthy men and women having similar plasma glucose levels, women exhibit enhanced plasma insulin and C-peptide concentrations after a meal [37], suggesting increased insulin secretion for a given glucose load in women. While our study did not measure insulin load, we have previously shown that sex differences in insulin secretion exist in mice at baseline [11]. The HFD significantly elevated blood insulin levels in male, but not age-matched female mice [11]; however, in this study, we did not measure post-prandial insulin to make a direct comparison. Here, we found that the stomach and the adipose tissue, but not the pancreas or the small intestine, of female mice display a higher uptake of FDG compared with male mice. While healthy women tend to have lower cardiovascular risk than men in general, a Finnish study found that diabetic women were three times more likely to die from coronary heart disease than diabetic men. Women with diabetes also tend to have more cardiovascular issues in general, such as impeded endothelium-dependent vasodilation and exacerbated atherogenic dyslipidemia [38].

A review of 621 studies involving 135,247 patients found that bariatric surgery improves diabetes in >85% of the diabetic population and remission of the disease in 78% [39]. However, causes of improvement or diabetes remission are not completely understood. In patients undergoing laparoscopic sleeve gastrectomy or the Roux-en-Y gastric bypass, diabetes remission or improvement occurs soon after surgery—well before significant weight loss. While reduced uptake and storage of fat in tissues is associated with bariatric surgery and changes in gut hormones; glucose uptake may also contribute.

## 5. Conclusions

In conclusion, post-prandial and fasting blood glucose levels are maintained in the normal range due to a combination of processes. These include actions of peptide hormones such as the insulin, glucagon, and somatostatin released primary from the pancreatic islets, as well as other gut peptide hormones, such as urocortins, ghrelin, leptin, etc., released from the gut. Together, these hormones, selectively and differentially influence glucose uptake in various tissues probably via distinct mechanisms. We reported that sex and diet are important independent variables that account for differential glucose uptake in brown fat, skeletal muscle, liver, heart, brain, kidney, perigonadal adipose, and the stomach, but not the lungs, pancreas, or the small intestine. The brown fat, perigonadal adipose and stomach tissues of young female mice had higher, whereas skeletal muscle and kidney tissues had lower FDG uptake than age-matched male mice. High-fat diet inhibited glucose uptake in three key organs that account for significant uptake on chow diet, whereas brain uptake was enhanced on high-fat diet in both sexes. It is important to note that, in this study, we did not investigate insulin-stimulated glucose uptake. The observed differential FDG uptake on high-fat diet maybe due to differential expression of GLUTs in various organs, specifically GLUT1. Diet-induced factors that are not insulin-dependent may also contribute to differential FDG uptake and warrant further study. Our seminal finding that brown fat is a key site for glucose homeostasis in young females is novel. The liver and heart appear to be key organs for glucose homeostasis in both sexes. Future studies that elucidate mechanisms of glucose homeostasis in diabetes, obesity, and metabolic syndrome must take into consideration that glucose uptake may not be equal in all organs and that sex is an important variable.

## Figures and Tables

**Figure 1 cells-11-02217-f001:**
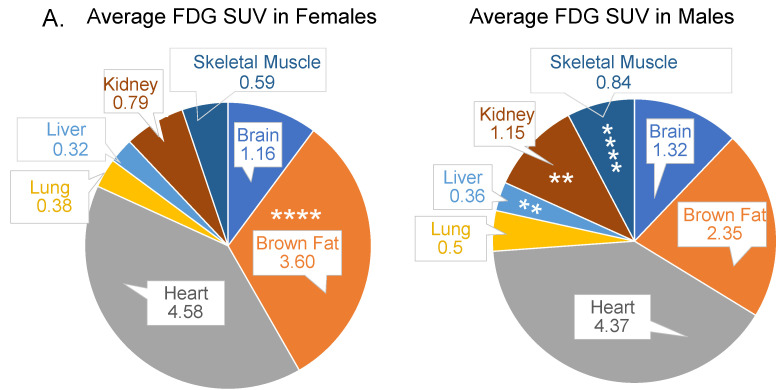
Differential FDG uptake by various organs. (**A**) Pie charts for average FDG SUVs in seven organs depicting that young female mice have greater FDG SUVs in the brown fat and lower FDG SUVs in the skeletal muscle, liver, and kidneys compared to age-matched male mice; the brain and heart SUVs were comparable. Significance as shown in bar graphs (**B**–**G**) Bar graphs showing sex as an independent significant variable in FDG uptake in several organs, except the brain and the heart, in young male and female mice on chow diet; *n* = 8–9/sex. Mann–Whitney *t*-test.

**Figure 2 cells-11-02217-f002:**
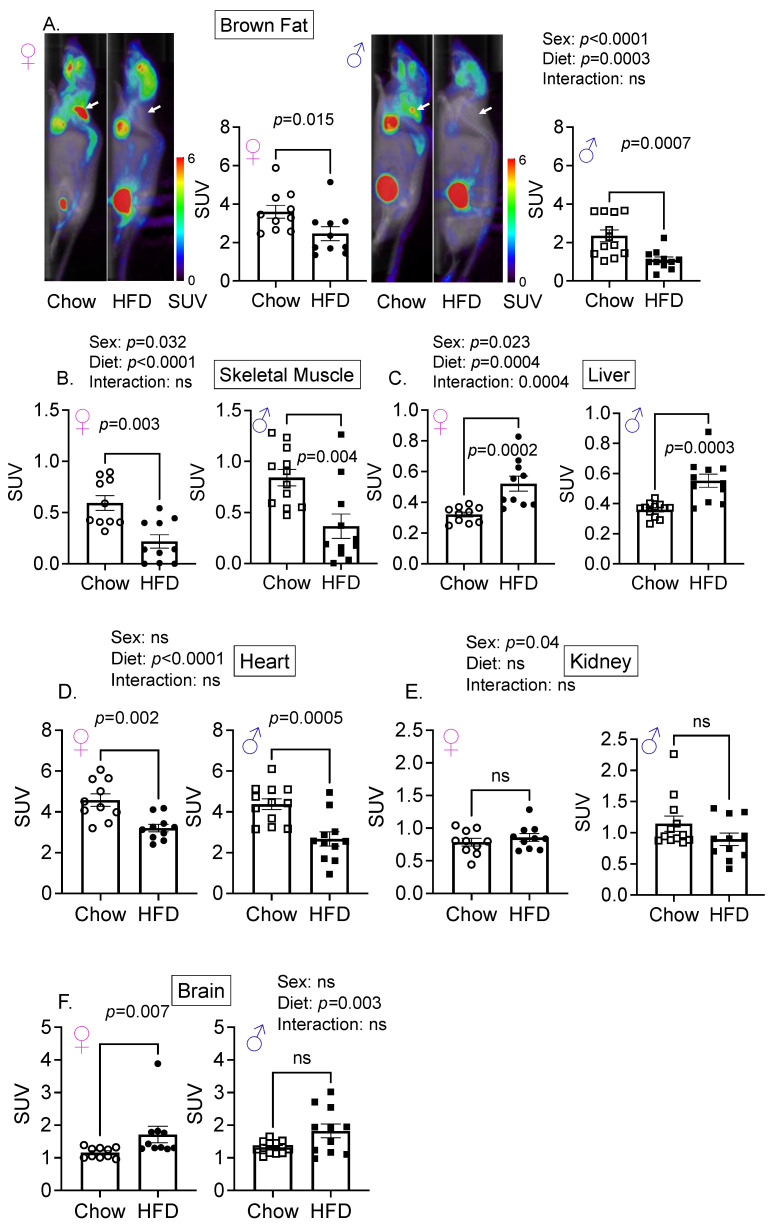
Sex and diet both influence FDG uptake in the skeletal muscle. (**A**) High-fat diet (HFD) decreased FDG uptake in the brown fat (arrows) and (**B**) skeletal muscle of both male and female mice. PET scans from representative male and female mice are shown. (**C**) HFD increased uptake in the livers, whereas (**D**) HFD decreased uptake in the heart of both sexes. (**E**) High-fat diet did not influence FDG uptake in the kidney in either sex or (**F**) in brains of male mice, but increased uptake in brains of female mice. (**G**) Pie charts depicting FDG SUVs in seven organs from which VOI was ascertained on chow and HFD in female and male young mice; *n* = 8–12/sex. Two-way ANOVA with sex and diet as independent variables was performed, followed by Mann–Whitney post hoc analysis. Please note that *y*-axis scale is different between the organs for better visualization of data points.

**Figure 3 cells-11-02217-f003:**
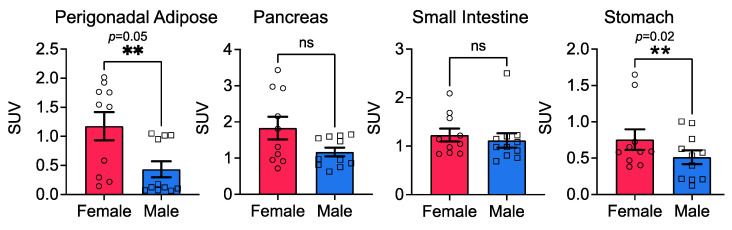
Sex and diet both influence FDG uptake in the stomach, but not in the pancreas, small intestine, or the perigonadal adipose; *n* = 10–16/sex. Two-way ANOVA was performed, followed by Mann–Whitney post hoc analysis.

**Figure 4 cells-11-02217-f004:**
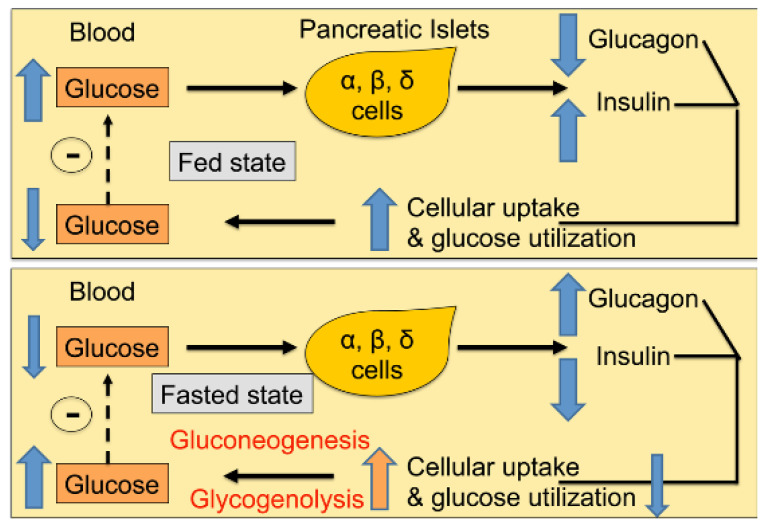
Glucose homeostasis is maintained by an interplay between glucose load; hormones secreted from the pancreas, the gut, and possibly adipose; and glucose uptake by key organs.

## Data Availability

Not applicable.

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
