# Peer review of "Organ-Specific Glucose Uptake: Does Sex Matter?"

_cells, 2022, doi:10.3390/cells11142217_

Round 1

Reviewer 1 Report

The present study fills a gap in the literature and highlights the importance of gender and diet effect in glucose uptake of several organs in mice, measured with PET-FDG

I have several comments:

- To validate further the methodology:

1)    Given that most of the VOI placed on the images are described as “spherical”, I would like to observe the VOI placements for the different organs in the three axes. The images do not have necessarily to be inserted in the main manuscript, could be in supplemental materials or just as an answer to my comment

2)    The authors have counted the organs, that could not be studied in vivo via PET after euthanasia. Did the authors measure counts also of the organs from which they obtain the SUV from the PET images? It would be important for the validation of the methodology to reassure the reader that the counted values ex-vivo are correlated and in the same range that the in-vivo, since the two type of measurements are put later in relation in Fig 1A (for example)

- There is missing a table or reference to the weight of the two gender groups, even if the SUV are normalized by weight, it is important to know if one group was heavier that the other, also in relation to the next point.

-Furthermore, did the HFD affect male and female differently, did they gain more or less weight? In this case, this situation might need extra corrections when analyzing the data statistically, maybe controlling for the delta increase in weight.

- I do not agree to refer to SUV measures as “glucose uptake”. Glucose uptake can be approximed/estimated with late PET scans (scans acquired some time after the injection of the radiotracer) if input function is measured via the calculation of fractional uptake rate (FUR). In this case the best the authors can do is to correct the SUV for glycemia (SUV gluc, named by someone). Fasting glycemia possible groups differences should be reported as well. If the glycemia is not available for this studies, I recommend refer to the measure as SUV or FDG uptake, but not glucose uptake. This pertains also the title. And discussion and conclusion should acknowledge the limitation that SUV from FDG PET is not a direct measurement of glucose uptake, given that the circulating glucose is not taking into account.

Minor comments:

- Fig 1 F, y axis label is VOI instead of SUV

- Some references in the discussion are: “PMID: XXXX”, so not properly formatted

Author Response

Reviewer 1

The present study fills a gap in the literature and highlights the importance of gender and diet effect in glucose uptake of several organs in mice, measured with PET-FDG

I have several comments:

- To validate further the methodology:

1) Given that most of the VOI placed on the images are described as “spherical”, I would like to observe the VOI placements for the different organs in the three axes. The images do not have necessarily to be inserted in the main manuscript, could be in supplemental materials or just as an answer to my comment

Response. Thank you for this comment. Please see sample image from each organ with VOIs, showing all three orthogonal axis images (transverse, coronal, and sagittal). These images are attached to the response, and we leave it up to the reviewer and/or the Editors whether these should be added to the supplemental materials.

2) The authors have counted the organs, that could not be studied in vivo via PET after euthanasia. Did the authors measure counts also of the organs from which they obtain the SUV from the PET images? It would be important for the validation of the methodology to reassure the reader that the counted values ex-vivo are correlated and in the same range that the in-vivo, since the two type of measurements are put later in relation in Fig 1A (for example)

Response. No, we did not measure counts from the organs from PET images. SUV from PET images is derived from the activity concentration in Bq/ml in reconstructed PET images. In our practice, we use a quantification calibration phantom that is a cylinder that has a known volume and known activity that is measured by a dose calibrator. After PET scan of this cylinder phantom, the value in the reconstructed images is calibrated to be converted to activity concentration by a multiplication factor. The gamma counter is also calibrated using an aliquot with known activity, to derive accurate activity. Hence, we could use activity measured by either instrument, which was calibrated against a known activity measured by a dose calibrator. Ours is not a technique paper and this is a standard technique used in human imaging as well. It is also a standard procedure to use organ counting instead of PET images for the stomach, pancreas, small intestine, and adipose tissue to obtain SUVs because these organs do not have precise locations and cannot be reliably determined without contrast. Thus, we used the PET scan images for the brain, lungs, heart, liver, kidneys, and skeletal muscle and gamma counts for the above-mentioned organs to obtain SUVs.

3) There is missing a table or reference to the weight of the two gender groups, even if the SUV are normalized by weight, it is important to know if one group was heavier that the other, also in relation to the next point.

Response. We now provide a graph (New Fig. S1) that shows average body weights taken on the day of imaging prior to injection of FDG. On both diets, males were heavier than females, on an average. It is a well-accepted fact that female rodents are less heavy than their male counterparts at all ages. Body weight change over days in C57BL6 mice on chow and HFD has also been described previously by us [1].

4) Furthermore, did the HFD affect male and female differently, did they gain more or less weight? In this case, this situation might need extra corrections when analyzing the data statistically, maybe controlling for the delta increase in weight.

Response. Body weights were measured (see new Table 1 and Fig. S1) to derive SUV that is normalized by the body weight. Hence, we consider that by using SUV, body weights are already incorporated as correction factors for our analysis.

5) I do not agree to refer to SUV measures as “glucose uptake”. Glucose uptake can be approximed/estimated with late PET scans (scans acquired some time after the injection of the radiotracer) if input function is measured via the calculation of fractional uptake rate (FUR). In this case the best the authors can do is to correct the SUV for glycemia (SUV gluc, named by someone). Fasting glycemia possible groups differences should be reported as well. If the glycemia is not available for this studies, I recommend refer to the measure as SUV or FDG uptake, but not glucose uptake. This pertains also the title. And discussion and conclusion should acknowledge the limitation that SUV from FDG PET is not a direct measurement of glucose uptake, given that the circulating glucose is not taking into account.

Response. We agree and thank the review for pointing this out. Glucose uptake has been changed to FDG uptake.

Minor comments:

- Fig 1 F, y axis label is VOI instead of SUV

- Some references in the discussion are: “PMID: XXXX”, so not properly formatted

Response thanks for pointing this oversight. The errors have been corrected.

Reference

  1. Paruthiyil, S.; Hagiwara, S.I.; Kundassery, K.; Bhargava, A. Sexually dimorphic metabolic responses mediated by CRF2 receptor during nutritional stress in mice. Biol Sex Differ 2018, 9, 49, doi:10.1186/s13293-018-0208-4.

Reviewer 2 Report

This is research article based on data obtained on animal model, about interesting theme, whether gender is an independent variable for differential glucose uptake by various organs. It has clinical importance, having in mind the great influence of these mechanisms on pathogenesis and therapy of type 2 diabetes, obesity and cardiovascular diseases. This is longitudinal study, and the authors used 18F-fluorodeoxyglucose (FDG) and positron emission tomography (PET) to determine baseline differences in whole body glucose uptake in young male and female mice on chow and high-fat diets. Brown fat, stomach, and heart tissues of young female mice had higher, whereas skeletal muscle and kidney tissues had lower glucose uptake than age-matched male mice. High-fat diet inhibited glucose uptake in most peripheral tissues examined, whereas brain uptake was enhanced in both sexes. The authors concluded that mechanisms of glucose homeostasis are context and organ-dependent and help explain sex-specific outcomes in treatment and management of diseases such as T2D, obesity, and metabolic syndrome. There are some issues to be clarified.

1.                   The manuscript should be proof-read by a native English speaker

2.                   Please, add clearly defined aim of the study in the abstract

3.                   Please, put some numeric values in the abstract

4.                   Please, reformulate the conclusion, emphasizing more your findings, and not general facts and data first mentioned at the end of the manuscript (lines 326-330)

Author Response

This is research article based on data obtained on animal model, about interesting theme, whether gender is an independent variable for differential glucose uptake by various organs. It has clinical importance, having in mind the great influence of these mechanisms on pathogenesis and therapy of type 2 diabetes, obesity and cardiovascular diseases. This is longitudinal study, and the authors used 18F-fluorodeoxyglucose (FDG) and positron emission tomography (PET) to determine baseline differences in whole body glucose uptake in young male and female mice on chow and high-fat diets. Brown fat, stomach, and heart tissues of young female mice had higher, whereas skeletal muscle and kidney tissues had lower glucose uptake than age-matched male mice. High-fat diet inhibited glucose uptake in most peripheral tissues examined, whereas brain uptake was enhanced in both sexes. The authors concluded that mechanisms of glucose homeostasis are context and organ-dependent and help explain sex-specific outcomes in treatment and management of diseases such as T2D, obesity, and metabolic syndrome. There are some issues to be clarified.

1)  The manuscript should be proof-read by a native English speaker

Response: Authors are native English speakers; we would appreciate if the reviewer can point to which sections do they think needs attention. However, we have corrected any typos and errors that might have remained.

2) Please, add clearly defined aim of the study in the abstract

Response: we have added a sentence outline the goal of this study in the abstract, as requested.

3) Please, put some numeric values in the abstract

Response: Thank you for this suggestion. Numeric values have been added as requested.

4) Please, reformulate the conclusion, emphasizing more your findings, and not general facts and data first mentioned at the end of the manuscript (lines 326-330)

Response: As suggested, we have added several sentences in the conclusion to re-emphasize our findings.

Reviewer 3 Report

Summary

Gandhi et al. determined basal glucose uptake into various organs and tissues in female and male mice fed a chow or HFD using 18F-fluorodeoxyglucose and positron emission tomography. They find diet and sex-dependent differences in basal glucose uptake into brown fat, skeletal muscle, liver, heart, kidney and the stomach. In particular, glucose uptake in chow-fed mice was lower in female compared to male mice in skeletal muscle, liver and kidney but increased in brown fat and the heart. Moreover, a HFD significantly increased glucose uptake in the brain (only in males) and the liver while it decreases it into brown fat and skeletal muscle. The authors conclude that observed differences in glucose uptake may explain sex-specific outcomes in treatment and management of diseases such as type 2 diabetes.

Broad comments

The finding that glucose uptake in several organs and tissues differs between sexes in chow-fed mice is interesting. To further understand involved mechanism, it would be helpful to know whether expression of different GLUT differs sex-dependently in individual organs. In contrast to observed sex-dependent differences in organ-specific glucose uptake in chow-fed mice, the effect of a HFD on organ-specific basal glucose uptake seems to be comparable between female and male mice (Figure 2), i.e. a HFD decreases glucose uptake in brown fat and skeletal muscle while increasing it in the liver. Hence, such similar effect of a HFD on basal glucose uptake questions the author’s conclusion that their findings may explain sex-specific outcomes in treatment and management of diseases such as type 2 diabetes. Moreover, since authors investigated herein basal but not insulin-stimulated glucose uptake, they should overall tone down the possible impact of their findings in the light of insulin resistance and type 2 diabetes. Clearly, a more comprehensive analysis including insulin-stimulated glucose uptake would be needed to draw such conclusion.

Specific comments

 It would be important to provide data of protein levels of different GLUT transporters in skeletal muscle, brown fat, liver, heart and kidney of male and female chow-fed mice. Such data would help to better interpret data presented in Figure 1.

Line 20: HFD inhibited glucose uptake in most peripheral tissues examined; this statement seems to be awkward, since HFD only reduced glucose uptake in two (brown fat, skeletal muscle) of six analyzed tissues.

Line 106: animals were fasted for at least 5 hours. What does that exactly mean? Where some animals fasted for longer durations? Since fasting may affect insulin levels, different fasting periods may impact on glucose uptake.

Line 154: uptake in eight major organs; since seven organs are listed afterwards, such sentence should be adapted.

Line 183: glucose uptake was x%; please define x

Line 185: in diabetes, obesity, and the metabolic syndrome…hepatic glucose uptake is reduced; please cite studies supporting such statement. Along the same line, authors show increased glucose uptake in livers of HFD-fed mice (Fig. 2C), which would suggest that glucose uptake may be rather increased in metabolically impaired situations.

Line 212: is glucose uptake into the stomach also higher in females compared to male mice under a chow diet?

Line 233: …with young female mice exhibiting higher glucose uptake than age-matched male mice; Looking at Fig. 2C, I cannot observe such regulation.

Line 235: please cite studies showing that all mentioned organs are able to perform gluconeogenesis.

Line 236: de novo glucose is synthesized via glycogenolysis; Glycogenolysis is the breakdown of stored glycogen to liberate glucose rather than de novo synthesis of glucose. 

Lines 308/309: please include References in a numerical way.

Author Response

Broad comments

The finding that glucose uptake in several organs and tissues differs between sexes in chow-fed mice is interesting. To further understand involved mechanism, it would be helpful to know whether expression of different GLUT differs sex-dependently in individual organs. In contrast to observed sex-dependent differences in organ-specific glucose uptake in chow-fed mice, the effect of a HFD on organ-specific basal glucose uptake seems to be comparable between female and male mice (Figure 2), i.e. a HFD decreases glucose uptake in brown fat and skeletal muscle while increasing it in the liver. Hence, such similar effect of a HFD on basal glucose uptake questions the author’s conclusion that their findings may explain sex-specific outcomes in treatment and management of diseases such as type 2 diabetes. Moreover, since authors investigated herein basal but not insulin-stimulated glucose uptake, they should overall tone down the possible impact of their findings in the light of insulin resistance and type 2 diabetes. Clearly, a more comprehensive analysis including insulin-stimulated glucose uptake would be needed to draw such conclusion. 

Response: We have removed the sentence “may explain sex-specific outcomes in treatment and management of diseases such as type 2 diabetes”. While we agree we did not give exogenous insulin, in response to food intake, insulin is released by the pancreas, thus glucose uptake in vivo is regulated by interplay between blood glucose and endogenous release of insulin and glucagon (Fig. 4).

Specific comments

 1) It would be important to provide data of protein levels of different GLUT transporters in skeletal muscle, brown fat, liver, heart and kidney of male and female chow-fed mice. Such data would help to better interpret data presented in Figure 1.

Response: Unfortunately, due to a terrible accident, our -80 freezer was found unplugged over 2 months ago and we have lost all our last 20 years of samples, including tissues from this study. Hence, at this point, it is not possible to provide this new data. While we agree that levels of different GLUT transporters in different tissues might help shed some light on this matter, but GLUT are not the only players, and whether there is any change (increase or decrease) or no change in the levels of GLUT transporters will not change the interpretation of the study. We have referenced work by others and us about changes in GLUT transporters and in absence of any other intervention, we do not believe that our findings will be any different than what is already known for expression of GLUTs at baseline.

2) Line 20: HFD inhibited glucose uptake in most peripheral tissues examined; this statement seems to be awkward, since HFD only reduced glucose uptake in two (brown fat, skeletal muscle) of six analyzed tissues.

Response: The sentence has been modified.

3) Line 106: animals were fasted for at least 5 hours. What does that exactly mean? Where some animals fasted for longer durations? Since fasting may affect insulin levels, different fasting periods may impact on glucose uptake.

Response: The sentence has been modified to at least 5-6 hours. Because of the size of the PET scan machine, only four mice could be scanned at a time and thus the scans had to be staggered. However, the difference in scan time between the first and last cages was one hour, and thus the maximum time the animals were fasted for was 6 hours (i.e. the mice in the first scan were fasted for five hours and the mice in the last scan were fasted for six hours).

4) Line 154: uptake in eight major organs; since seven organs are listed afterwards, such sentence should be adapted. 

 Response: The sentence has been modified accordingly.

5) Line 183: glucose uptake was x%; please define x

Response: Apologies for this oversight. X (47%) has been defined.

6) Line 185: in diabetes, obesity, and the metabolic syndrome…hepatic glucose uptake is reduced; please cite studies supporting such statement. Along the same line, authors show increased glucose uptake in livers of HFD-fed mice (Fig. 2C), which would suggest that glucose uptake may be rather increased in metabolically impaired situations.

Response: As shown in the studies cited (references 19-21), uptake is reduced in diabetics. However, it is important to note that the mice in the study were not yet diabetic after 2 weeks on HFD; the study design was to see how short-term changes in diet changes that might predispose one to diabetes might influence glucose uptake. Thus, it is possible that uptake initially increases in response to the change in diet to make up for the decrease in sensitivity to insulin.

7) Line 212: is glucose uptake into the stomach also higher in females compared to male mice under a chow diet?

Response:  Because this was a longitudinal study, we could not obtain counting data for the stomach in mice on chow diet (part 1 of the study). Additionally, without contrast we could not reliably obtain data for the glucose uptake in stomach using the PET scans. Thus, we could not investigate sex differences in uptake in the stomach for mice under a chow diet.

8) Line 233: …with young female mice exhibiting higher glucose uptake than age-matched male mice; Looking at Fig. 2C, I cannot observe such regulation.

Response: We conducted a two-way ANOVA looking at sex and diet. Combining the results for both diets, the average glucose uptake in females was higher than in males (p = 0.02). The figure separates the results by diet and thus does not show this difference.

9) Line 235: please cite studies showing that all mentioned organs are able to perform gluconeogenesis.

Response: Studies have been cited for organs that have been shown to perform gluconeogenesis (new references 24-27).

10) Line 236: de novo glucose is synthesized via glycogenolysis; Glycogenolysis is the breakdown of stored glycogen to liberate glucose rather than de novo synthesis of glucose.  

Response: We thank the reviewer for catching this. The sentence has been modified.

11) Lines 308/309: please include References in a numerical way.

Response. The references have been formatted.

Round 2

Reviewer 1 Report

Most of my comments have been addressed.

1)I have only a concern about the placement of the heart VOI.

In the additional material presented the VOI is placed almost exactly in the LV (left ventricle) that it is usually the region used for extracting the input function in the quantitative dynamic PET studies. I would prefer that a VOI would be placed on the myocardium (the high activity "ring", around the levf ventricle), in order to reflect the FDG uptake in the heart muscle. The SUVs will result higher as expected.

2) As a suggestion (not mandatory). In Fig 1, I would put B, D, G in one row with the same scale 0 - 2.5; and C, E, F with appropriate scale (after heart recalculation) 0 - 8 (at the moment) in order to easily compare the different organs at a quick glance.

Author Response

1)I have only a concern about the placement of the heart VOI.

In the additional material presented the VOI is placed almost exactly in the LV (left ventricle) that it is usually the region used for extracting the input function in the quantitative dynamic PET studies. I would prefer that a VOI would be placed on the myocardium (the high activity "ring", around the levf ventricle), in order to reflect the FDG uptake in the heart muscle. The SUVs will result higher as expected.

Response. We thank the reviewer for this suggestion. We -re-measured FDG uptake by placing a thin box on the myocardium, as suggested. An elliptic cylinder of dimension 1.5 mm short and long axes, and 3 mm height over the muscle was used. The new data shows no difference in FDG uptake between the sexes on chow diet, whereas on HFD uptake was decreased in both sexes. We have revised all pertinent figures accordingly.

2) As a suggestion (not mandatory). In Fig 1, I would put B, D, G in one row with the same scale 0 - 2.5; and C, E, F with appropriate scale (after heart recalculation) 0 - 8 (at the moment) in order to easily compare the different organs at a quick glance.

Response. Figure 1B-G have been re-arranged and Y-axis scale for organs have been adjusted so that the scales are identical between the sexes for the same organ. But since uptake between organs is highly variable, we have used a scale that does not compromise visualization of data. Hope these changes have addressed the concerns.

Reviewer 3 Report

Authors have adequately addressed most of my concerns. However, I would still ask to tone down the possible impact of their findings in the light of insulin resistance since they did not investigate insulin-stimulated glucose uptake. Although I am aware that insulin is released by the pancreas, and, thus affects glucose uptake also in the basal state, it may be that differences in the expression of (non-insulin dependent) expression of GLUT1 critically contributed to observed effects in HFD-fed animals.

Author Response

Reviewer 3:

Authors have adequately addressed most of my concerns. However, I would still ask to tone down the possible impact of their findings in the light of insulin resistance since they did not investigate insulin-stimulated glucose uptake. Although I am aware that insulin is released by the pancreas, and, thus affects glucose uptake also in the basal state, it may be that differences in the expression of (non-insulin dependent) expression of GLUT1 critically contributed to observed effects in HFD-fed animals.

Response. We appreciate the reviewer’s concern. We have now mentioned this caveat in our discussion (Lines 301-302) and modified our conclusion to include:

It is important to note that in this study, we did not investigate insulin-stimulated glucose uptake. On high-fat diet, differential expression of GLUTs in various organs, specifically GLUT1 due to diet-induced factors that are not insulin-dependent, may contribute to differential FDG uptake and warrant further study. Hope these changes have addressed the concerns.